# Rhodoquinone biosynthesis in *C. elegans* requires precursors generated by the kynurenine pathway

Samantha Del Borrello[1†], Margot Lautens[1†], Kathleen Dolan[1†], June H Tan[1], Taylor Davie[1], Michael R Schertzberg[1], Mark A Spensley[1,2]*, Amy A Caudy[1]*, Andrew G Fraser[1]*

[1]The Donnelly Centre, University of Toronto, Toronto, Canada; [2]Whole Animal Phenotyping, Phenalysys Inc, Toronto, Canada

**Abstract** Parasitic helminths infect over a billion humans. To survive in the low oxygen environment of their hosts, these parasites use unusual anaerobic metabolism — this requires rhodoquinone (RQ), an electron carrier that is made by very few animal species. Crucially RQ is not made or used by any parasitic hosts and RQ synthesis is thus an ideal target for anthelmintics. However, little is known about how RQ is made and no drugs are known to block RQ synthesis. *C. elegans* makes RQ and can use RQ-dependent metabolic pathways — here, we use *C. elegans* genetics to show that tryptophan degradation via the kynurenine pathway is required to generate the key amine-containing precursors for RQ synthesis. We show that *C. elegans* requires RQ for survival in hypoxic conditions and, finally, we establish a high throughput assay for drugs that block RQ-dependent metabolism. This may drive the development of a new class of anthelmintic drugs. This study is a key first step in understanding how RQ is made in parasitic helminths.
DOI: https://doi.org/10.7554/eLife.48165.001

*For correspondence:
maspensley@gmail.com (MAS);
amy.caudy@utoronto.ca (AAC);
andyfraser.utoronto@gmail.com
(AGF)

[†]These authors contributed
equally to this work

Competing interest: See
page 18

Reviewing editor: Phillip A
Newmark, HHMI/Morgridge
Institute for Research, University
of Wisconsin-Madison, United
States

## Introduction

Soil-transmitted Helminths (STHs) are major human pathogens (*WHO Expert Committee on the Control of Schistosomiasis, 2002*). Over a billion humans are infected with an STH — the roundworm *Ascaris lumbricoides,* the whipworm *Trichuris trichuria*, and the hookworm *Necator americanus* account for most of these infections (*WHO Expert Committee on the Control of Schistosomiasis, 2002*). STHs are transmitted from human to human via the soil where eggs from human faeces develop into infective stages which then enter new hosts (reviewed in *Brooker et al., 2006*). On infection, STHs encounter a very different environment and require multiple strategies to be able to survive. One of the major changes is the availability of oxygen. While there is abundant oxygen outside their hosts, in many host tissues there is little available oxygen — this is especially true in the intestine where oxygen levels drop steeply to near anoxia in the lumen (reviewed in *Espey, 2013*). To survive in these hypoxic conditions, parasites must switch from aerobic respiration to anaerobic respiration; crucially, the anaerobic metabolic pathways that STHs depend on are unusual and are not used in any host (*Van Hellemond et al., 1995*). Inhibiting these anaerobic pathways is therefore a possible way to kill the parasites while leaving the host unaffected.

During aerobic respiration in helminths, the great majority of ATP is made in the mitochondrion (*Tielens, 1994*; *Tielens et al., 1984*). Electrons enter the Electron Transport Chain (ETC) either at Complex I or via several quinone-coupled dehydrogenases (QDHs from here on). These QDHs include Succinate Dehydrogenase (Complex II) and Electron-Transferring Flavoprotein Dehydrogenase (ETFDH) (*Komuniecki et al., 1989*; *Ma et al., 1993*; *Rioux and Komuniecki, 1984*). The electrons entering the ETC are first transferred to the lipid soluble electron carrier ubiquinone (UQ)

**eLife digest** Parasitic worms infect more than a billion people worldwide, using a range of tricks to survive inside the human body. Some species can live for weeks inside the gut, a place with almost no oxygen. Yet exactly how they manage this is remains unclear.

Scientists know that parasitic worms have an unusual way of making chemical energy when oxygen levels drop. Like human cells, worm cells use a series of molecular complexes called the electron transport chain. As electrons pass along the chain, they drive the production of chemical energy. Normally, oxygen sits at the end of the chain to receive the electrons. But, when there is no oxygen, almost all animals stop using the electron transport chain. A few animals can continue to use it by using other molecules to receive the final electrons instead of oxygen. To do that, they need a special electron carrier and, in worms, this electron carrier is rhodoquinone.

Human cells do not use rhodoquinone, making it a prime target for drug design. If a drug could block rhodoquinone production, it might be able to stop worms surviving in the human intestines without harming the patient's own cells. Yet, even though the scientific community has known about rhodoquinone for more than 50 years, it remains unclear how worms make this molecule.

To find out, Del Borrello et al. examined the laboratory worm *Caenorhabditis elegans*. This worm is not a parasite, but it does make rhodoquinone. Del Borrello et al. developed a new way to study rhodoquinone production by blocking the normal route of the electron transport chain with cyanide. This causes the worms to switch to using rhodoquinone and is cheaper than raising the worms in low oxygen, making it easier to conduct high-throughput screening. A combination of chemistry and information from other species made it possible to identify candidate genes responsible for the production of rhodoquinone. Worms with faults in these genes revealed the key building blocks of rhodoquinone, and the early steps in its production. Removing any one of the genes made it harder for the worms to survive without oxygen.

Although there are already effective drugs that kill parasitic worms, resistance is growing. A better understanding of rhodoquinone could lead to a new class of drugs to help control this major problem in global health. A drug that blocks any one of the production steps of rhodoquinone might be a future candidate for a new anti-parasitic worm therapy.

DOI: https://doi.org/10.7554/eLife.48165.002

(*Crane et al., 1957*; *Mitchell, 1975*). From UQ, they are ultimately carried to Complex III then IV where they are finally transferred onto oxygen as the terminal electron acceptor (see *Figure 1a*). Electron transport is coupled to proton pumping into the inner membrane space of the mitochondrion — this establishes a proton gradient which is used to power the F0F1-ATP synthase (*Mitchell, 1961*). When there is insufficient oxygen to accept electrons at Complex IV, or when inhibitors of Complex IV such as cyanide are present (*Antonini et al., 1971*; *Nicholls et al., 1972*), almost all animals stop using the ETC and rely on anaerobic glycolysis to make ATP, generating lactate (*Isom et al., 1975*; *Meyerhof, 1927*). STHs, however, have evolved a different solution that allow them to survive months in the hypoxic host environment. Electrons still enter the ETC at Complex I, Complex I still pumps protons to generate the proton motive force (PMF), and ATP is still made by the F0F1ATPase, powered by the PMF. However, rather than the electrons passing through the ETC to oxygen as the terminal electron acceptor, they exit the ETC immediately downstream of Complex I onto a number of alternative terminal electron acceptors (*Figure 1b*) (reviewed in *Hochachka and Mustafa, 1972*; *Müller et al., 2012*). This transfer of the electrons out of the ETC and onto alternative electron acceptors requires the quinone-coupled dehydrogenases (*Kita, 1992*; *Ma et al., 1993*). Under aerobic conditions these QDHs act as entry points to the ETC, transferring electrons from their substrates to UQ. Crucially, the reactions catalysed by these QDHs are reversed in anaerobic conditions — they now act as reductases transferring electrons out of the ETC and onto their products. For example, Complex II acts as a succinate dehydrogenase in aerobic conditions, generating fumarate; in anaerobic conditions, it reduces fumarate generating succinate as a terminal electron sink (*Figure 1c*) (*Kmetec and Bueding, 1961*; *Sato et al., 1972*; *Saz and Vidrine, 1959*; *Van Hellemond et al., 1995*; *Takamiya et al., 1999*). In this way, an entry of electrons into the ETC

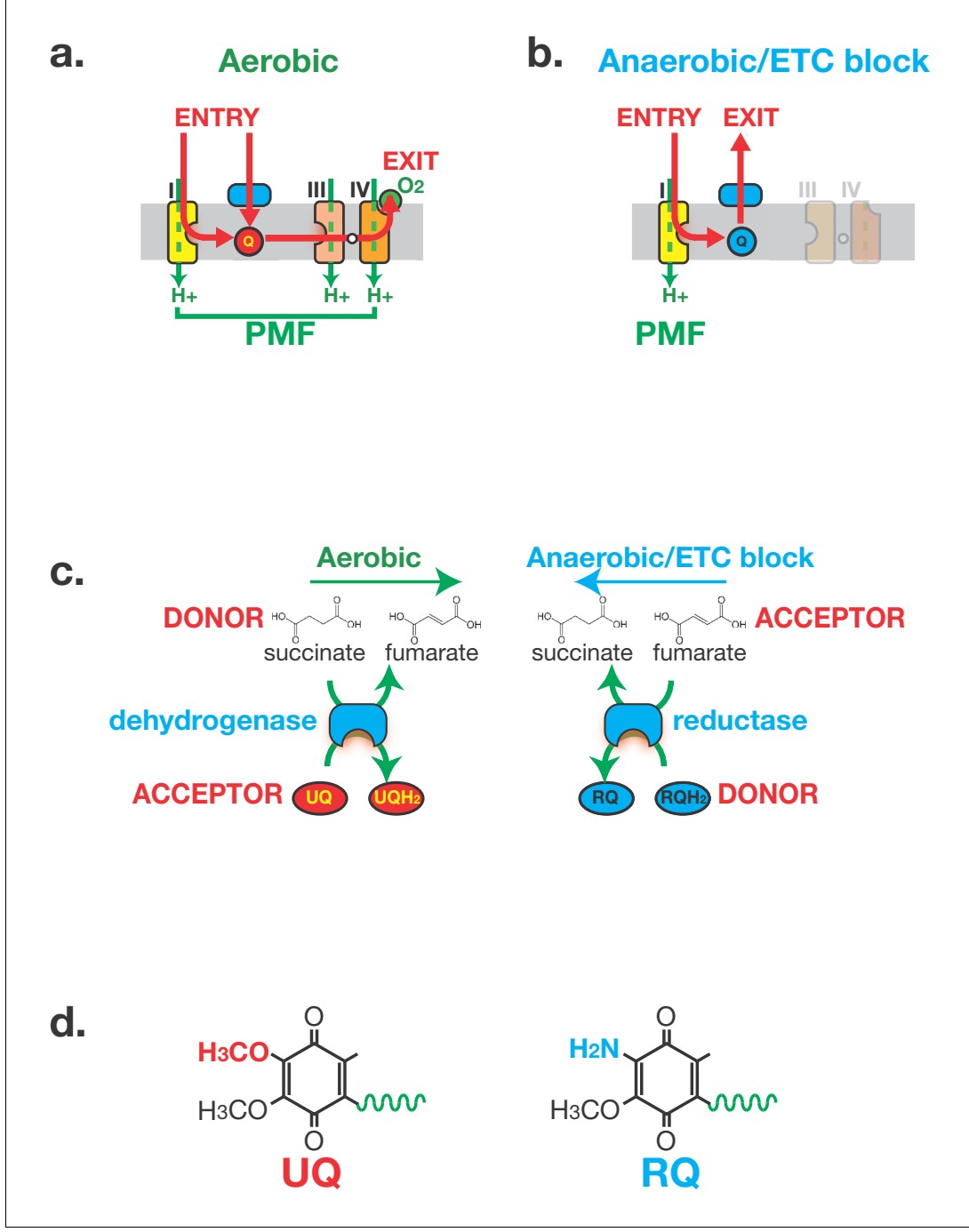

**Figure 1.** Anaerobic metabolism in helminths requires Rhodoquinone (RQ). (**a**) Electron flow in the Electron Transport Chain (ETC) under aerobic conditions. Electrons enter the ETC either via Complex I or via a number of Quinone-coupled Dehydrogenases (QDH; cyan). These complexes transfer electrons to Ubiquinone (red circle 'UQ') which shuttles them to Complex III. They exit the ETC at Complex IV where they are transferred to oxygen as the terminal electron acceptor. Proton pumping is coupled to electron transport and is carried out by Complexes I, III and IV. Electron flow is shown in red and proton pumping in green. (**b**) Electron flow in the Electron Transport Chain (ETC) under anaerobic conditions. Electrons still enter the ETC at Complex I which transfers electrons to RQ (cyan circle 'RQ'). RQ shuttles electrons to the QDHs which now operate as reductases, allowing electrons to exit the ETC and onto a diverse set of terminal electron acceptors. Complex I is the sole proton pump in this truncated ETC. (**c**) Schematic of Complex II activity under aerobic and anaerobic conditions. Under aerobic conditions, Complex II acts as a succinate dehydrogenase, transferring electrons from succinate onto UQ. Under anaerobic

*Figure 1 continued on next page*

*Figure 1 continued*

conditions, Complex II operates in the reverse direction acting as a fumarate reductase, accepting electrons from RQ and transferring them to succinate as the terminal electron sink. (**d**) Structure of UQ and RQ. The critical amine group differing between UQ and RQ is highlighted; the polyprenyl tail is shown schematically as a green wavy line.
DOI: https://doi.org/10.7554/eLife.48165.003

from a variety of electron donors in aerobic conditions is reversed to provide an exit from the ETC onto a variety of electron acceptors in anaerobic conditions.

The unusual ETC wiring used by STHs to survive anaerobic conditions requires an unusual electron carrier, rhodoquinone (RQ) (*Moore and Folkers, 1965*). RQ and UQ are highly related molecules — the sole difference is the presence of an amine group on the quinone ring of RQ (*Figure 1d*). This changes the biophysical properties of the quinone ring: while UQ can accept electrons from the QDHs as they flow into the ETC under aerobic conditions, UQ cannot carry electrons of the correct electropotential to drive the reverse reactions in anaerobic conditions (*Fioravanti and Kim, 1988*; *Sato et al., 1972*). RQ can carry such electrons (*Fioravanti and Kim, 1988*; *Sato et al., 1972*), however, and the ability of STHs to drive their unusual anaerobic metabolism in their hosts is absolutely dependent on RQ. The single amine group that differs between UQ and RQ thus affects the health of over a billion humans.

RQ is found in very few animal species — only helminths, molluscs and annelids are known to make RQ (*Allen, 1973*; *Fioravanti and Kim, 1988*; *Sato and Ozawa, 1969*; *Takamiya et al., 1999*; *Van Hellemond et al., 1995*). Since no host animals make RQ, inhibiting RQ synthesis or RQ use is a potentially powerful way to target parasites inside their host. Currently however little is known about RQ synthesis. The most mature studies have focused on the purple Proteobacterium *R.rubrum*, where RQ appears to derive from UQ (*Brajcich et al., 2010*). RQ synthesis in *R.rubrum* requires the gene *rquA* (*Lonjers et al., 2012*) which is the first and thus far only gene known to be required for RQ synthesis in any organism. The role of *rquA* was initially unclear (*Lonjers et al., 2012*), but very recently it has emerged that it is capable of converting UQ to RQ (*Bernert et al., 2019*). In animals, the situation is even more blank: nothing is known about which genes are required for RQ synthesis, there are no clear *rquA* orthologues (*Lonjers et al., 2012*), and there are no drugs that are known to prevent RQ synthesis. This is partly because no tractable animal model has been established in which to study RQ synthesis and use.

Previous studies have shown that *C. elegans*, a free-living helminth, can make RQ (*Takamiya et al., 1999*) and that when *C. elegans* is exposed to hypoxic conditions it undergoes major metabolic changes that resemble those that occur in STHs when they are in the hypoxic environment of their hosts (*Butler et al., 2012*; *Föll et al., 1999*). This suggested to us that we could establish *C. elegans* as a model for dissecting the pathway of RQ synthesis and for screening for drugs that block RQ synthesis or use. We confirm that *C. elegans* makes RQ and also that it uses RQ-dependent metabolism when unable to use oxygen as a terminal electron acceptor. We show that in *C. elegans* RQ synthesis requires the activity of the kynurenine pathway. This pathway is key for the metabolism of tryptophan and generating metabolites that include 3-hydroxyanthranilate (3 HA). Our data suggest that 3 HA (or a highly related molecule generated by the kynurenine pathway) is the amine-containing precursor for RQ and that it is the source of the critical amine group on the quinone ring of RQ. Thus in helminths the amine group is present from the start of the RQ biosynthetic pathway and is not added as a late step to UQ as is the case in bacteria. We also show that *C. elegans* requires RQ to survive under conditions where oxygen cannot be used as an electron acceptor for the ETC. This allowed us to establish a high throughput screening assay to identify compounds that block RQ synthesis or RQ use. This is the first study to show how RQ, an electron carrier that affects the life of over a billion humans, is made in helminths. This will help towards the development of a new class of drugs to treat these major human pathogens.

## Results

### *C. elegans* makes RQ and switches to RQ-dependent metabolism when exposed to potassium cyanide

*C. elegans* is a non-parasitic helminth that is easily genetically tractable (*Brenner, 1974*; reviewed in *Jones et al., 2005*) and can be used for efficient drug screens (*Burns et al., 2015*). We wanted to establish *C. elegans* as a model to dissect the pathway for RQ synthesis in helminths and as a system in which we could efficiently screen for drugs that block the synthesis of RQ or use of RQ. We note that there are no other standard model organisms where this is possible: yeasts, insects, fish, and vertebrates do not make or use RQ so *C. elegans* is the sole genetically tractable animal model for this work.

Like other helminths, *C. elegans* has previously been shown to make RQ (*Takamiya et al., 1999*). We wanted to confirm this and determine whether we could define a simple experimental method to drive *C. elegans* to carry out similar RQ-dependent anaerobic metabolism as that used by parasitic helminths to survive in their hosts. We extracted quinones as described in Materials and methods and as shown in *Figure 2a* and Supp *Figure 1*, *C. elegans* makes both UQ and RQ when maintained in normoxic conditions. We can therefore use *C. elegans* to genetically dissect the pathway for RQ synthesis.

Our next step was to establish a simple method to drive *C. elegans* to use RQ-dependent metabolism that would allow high throughput drug screens. Parasitic helminths use RQ-dependent metabolism under low oxygen conditions (*Rioux and Komuniecki, 1984*; *Saz and Lescure, 1969*; *Tielens et al., 1992*) and previous studies showed that *C. elegans* shows similar metabolic shifts when exposed to hypoxic conditions (*Butler et al., 2012*; *Föll et al., 1999*; ). If possible, however,

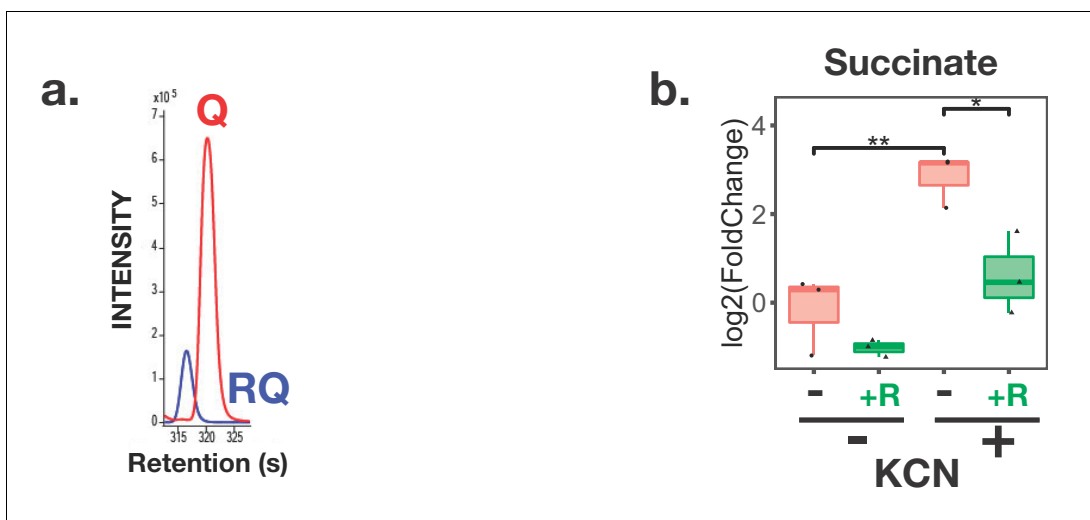

**Figure 2.** *C. elegans* makes Rhodoquinone (RQ) and can carry out RQ-dependent anaerobic metabolism. (**a**) *C. elegans* makes both UQ and RQ. *C. elegans* were grown under normoxic conditions and quinones extracted and analysed by mass spectrometry (see Materials and methods). Both UQ and RQ can be detected. (**b**) *C. elegans* increases succinate production following treatment with Potassium Cyanide (KCN). *C. elegans* L1 larvae were treated either with 200 µM KCN alone, 12.5 µM rotenone alone, or a combination of KCN and rotenone for 1 hr and metabolites extracted and analysed by mass spectrometry (see Materials and methods). The graph shows that succinate levels increase over 5-fold following KCN treatment and that this increase is blocked by rotenone indicating that it requires Complex I activity. Data are from three independent biological repeats; box plots show median and the interquartile range as well as individual data points; succinate levels are shown relative to levels in wild-type untreated animals.

DOI: https://doi.org/10.7554/eLife.48165.004

The following figure supplement is available for figure 2:

**Figure supplement 1.** Mass spectra for Rhodoquinone (RQ) and Ubiquinone (UQ).

DOI: https://doi.org/10.7554/eLife.48165.005

we wanted to avoid the use of hypoxic chambers. While hypoxia chambers are highly accurate ways of controlling oxygen levels, they are also very expensive and cumbersome for large-scale drug screens. We therefore turned to chemical methods of inducing a hypoxic state. Potassium Cyanide (KCN) is a potent inhibitor of Complex IV (*Antonini et al., 1971*; *Nicholls et al., 1972*) — KCN inhibits oxygen binding to Complex IV and KCN treatment thus mimics the effect of hypoxia on the ETC. We tested whether treatment with KCN could drive *C. elegans* to use anaerobic metabolism that is similar to the RQ-dependent metabolism used by STHs in their hosts. The classic hallmark of RQ-dependent anaerobic metabolism in helminths is the generation of high levels of succinate through the reversal of Complex II (*Figure 1c*) (*Butler et al., 2012*; *Saz and Lescure, 1969*; *Tielens et al., 1992*). If *C. elegans* can indeed use the same anaerobic metabolism as parasitic helminths, there should be a build-up of succinate following KCN treatment. Furthermore, this should be dependent on Complex I activity, since Complex I is the sole source of electrons that are carried by RQ to drive the fumarate reductase activity of Complex II (*Figure 1b*). We found that when *C. elegans* are exposed to KCN, they build up high levels of succinate as expected and that inhibiting Complex I with rotenone prevents succinate build-up (*Figure 2b*). We thus find that *C. elegans* makes RQ and that treatment of *C. elegans* with KCN causes them to switch to a metabolic state that resembles that of STHs in their host. *C. elegans* is thus an excellent model in which to dissect RQ synthesis and to screen for compounds that alter RQ-dependent metabolism.

## In *C. elegans*, RQ is synthesised from products of the kynurenine pathway and not from ubiquinone

RQ and UQ are highly related molecules — the sole difference is the presence of an amine group on the quinone ring of RQ (*Figure 1d*). A critical question for RQ synthesis is where this amine group comes from and how it is generated. The best-defined current model for RQ synthesis comes from experiments in the proteobacterium *R.rubrum*. At least in this prokaryote, RQ is thought to be made by a late addition of the critical amine group to an existing molecule of UQ (*Brajcich et al., 2010*). UQ is thus an obligate precursor of RQ and RQ synthesis requires initial synthesis of UQ (*Figure 3a*). While this may be the case for *R.rubrum*, this is not the case in *F.hepatica* (*Van Hellemond et al., 1996*) or *C. elegans*. The *clk-1(qm30)* strain has a loss-of-function mutation in the *C. elegans* COQ7 orthologue that is required for hydroxylation of 5-demethoxyubiquinone to 5-hydroxyubiquinone, a late step in UQ synthesis — there is no detectable UQ in *clk-1(qm30)* homozygous animals. However, a previous study showed that there does appear to be RQ in this strain (*Jonassen et al., 2001*). If there is no UQ, but there is RQ, then RQ is not derived from UQ, at least in helminths. The two models for RQ synthesis thus differ fundamentally — in one UQ is an obligate precursor (*Brajcich et al., 2010*), in the other it is not (*Jonassen et al., 2001*). Since this is a fundamental result, we wanted to confirm this before trying to dissect the pathway of RQ synthesis. We thus extracted and analysed quinones from either wild-type worms or *clk-1(qm30)* homozygous animals. We find that while there is no detectable UQ in *clk-1(qm30)* mutants, there is abundant RQ and indeed we find that RQ levels are essentially unchanged (*Figure 3b*). We thus confirm that UQ is not an obligate precursor for RQ in *C. elegans*.

If RQ is not generated by addition of the key amine group to an existing UQ molecule where does the amine group on RQ come from? One possibility is it is added not to UQ but to a UQ precursor such as demethoxyquinone (DMQ) — such UQ precursors would still be present in the *clk-1 (qm30)* mutant strain (*Figure 3a* for schematic; DMQ is abundant in *clk-1(qm30)* (data not shown)). While this could in principle be the case, it is unlikely because amination of an aromatic ring is highly thermodynamically unfavourable (reviewed in *Downing et al., 1997*). We therefore investigated an alternative possibility — that the critical amine group of RQ is not added in a late step of RQ synthesis but instead is present from the outset.

A key initial step in UQ synthesis is the addition by COQ-2 of a polyprenyl tail to a *p*-hydroxybenzoate ring (PHB — also often called 4-hydroxybenzoate (4-HB)) (*Momose and Rudney, 1972*; *Trumpower et al., 1974*). PHB has no amine group — however, *S. cerevisiae* COQ2 is known to be able to use a variety of similar compounds as substrates for prenylation such as para-aminobenzoic acid and vanillic acid (reviewed in *Pierrel, 2017*). Given the potential substrate flexibility of COQ-2, we hypothesised that RQ synthesis might start not with PHB but with a related molecule that contains an amine group already on the ring (*Figure 3a* for schematic). In particular, we noted that yeast COQ2 is tolerant of substituents at positions 5 and 6 of the PHB structure (reviewed in *Pierrel, 2017*)

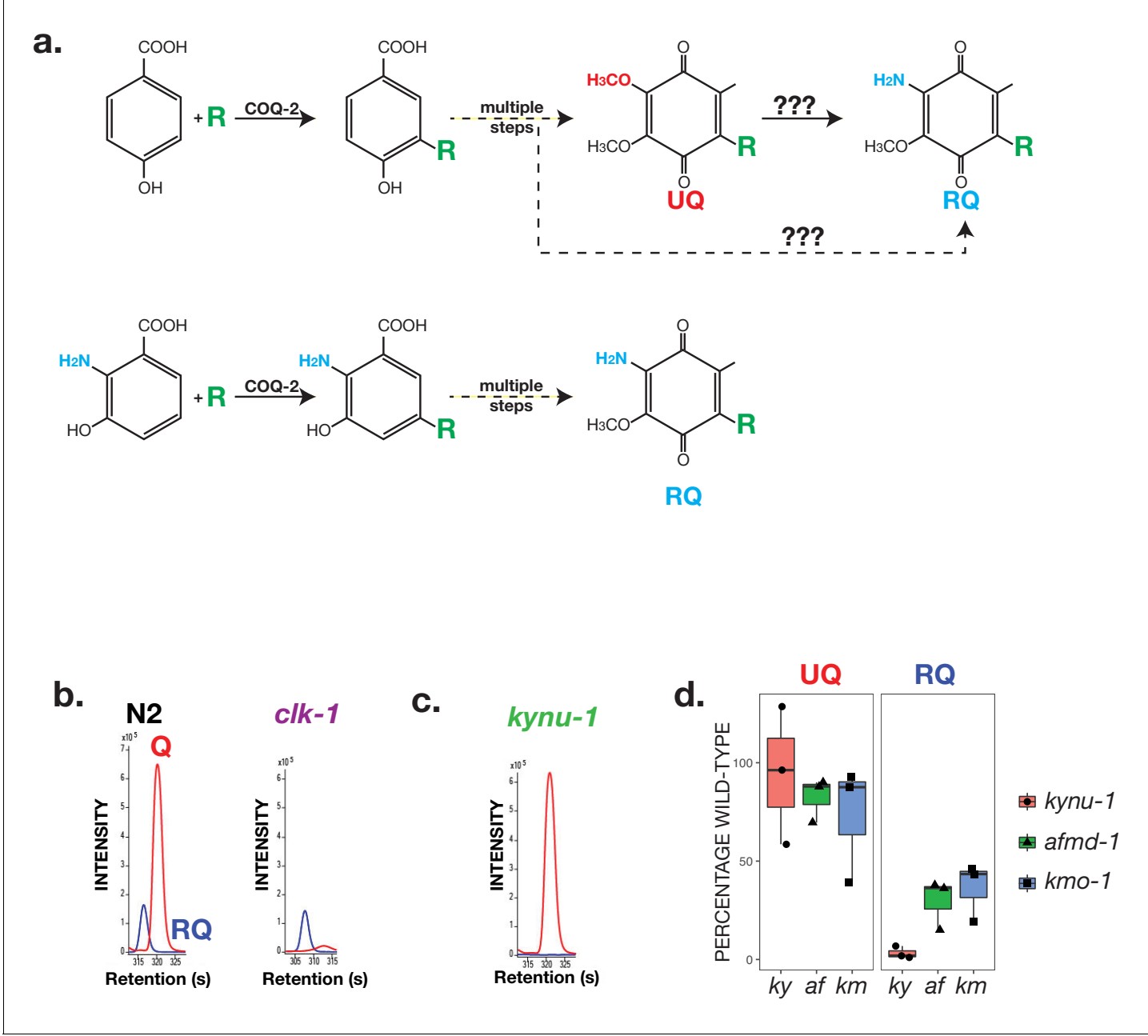

**Figure 3.** RQ in *C. elegans* does not derive from UQ but from Tryptophan metabolites. (a) Schematic showing possible routes for RQ synthesis. Current models for RQ synthesis are shown schematically in the top pathway: PHB is prenylated by COQ-2 at the start of the UQ synthesis pathway and RQ either derives from UQ or from a UQ precursor — the amine group is thus added at a late step. The lower pathway shows our proposed pathway. Rather than use PHB as the substrate for COQ-2, RQ synthesis starts with prenylation of 3-hydroxyanthranilate by COQ-2. The amine group is thus present from the start of RQ synthesis rather than being added at a late step and RQ synthesis proceeds via the same pathway as UQ synthesis. (b) UQ is not a required intermediate for RQ synthesis. *clk-1(qm30)* mutant worms and N2 wild-type worms were grown under normoxic conditions and quinones extracted and analysed. N2 worms contain both UQ and RQ whereas *clk-1(qm30)* mutants only contain detectable RQ. (c) RQ synthesis requires metabolism of Tryptophan via the kynurenine pathway. *kynu-1(e1003)* mutant worms were grown under normoxic conditions and quinones extracted and analysed. While wild-type worms contain both UQ and RQ, *kynu-1(e1003)* mutants only contain detectable UQ. RQ synthesis thus requires the degradation of Tryptophan via the kynurenine pathway. (d) UQ and RQ levels in different kynurenine pathway mutants. Quinones were extracted from worms homozygous for loss of function mutations in *kynu-1*, *afmd-1* or *kmo-1* (*ky*, *af* and *km* respectively) and analysed by mass spec. Graphs show levels of UQ and RQ in each strain — levels are shown as % of wild-type levels, and each data point derives from a separate biological replicate.

DOI: https://doi.org/10.7554/eLife.48165.006

*Figure 3 continued on next page*

*Figure 3 continued*

The following figure supplement is available for figure 3:

**Figure supplement 1.** Extracted ion chromatogram and possible structure of suspected RQ intermediate likely to polyprenyl 3-hydroxyanthranilate (m/z = 768.63).

DOI: https://doi.org/10.7554/eLife.48165.007

suggesting that this might be feasible enzymatically. We considered different candidate molecules as amine-containing ring structures that might act as precursors for RQ and focussed on anthranilate and 3-hydroxyanthranilate (3 HA) as likely sources of the amine-containing ring in RQ. Anthranilate and 3 HA are made from the amino acid tryptophan via the kynurenine pathway (*Heidelberger and Gullberg, 1948*; *Kotake, 1936*) and *kynu-1* encodes the *C. elegans* kynureninase that is required for the generation of anthranilate and 3 HA (*Babu, 1974*; *Bhat and Babu, 1980*; *van der Goot et al., 2012*). We examined the quinones present in *kynu-1(e1003)* mutants that lack kynureninase and while UQ levels are normal in the *kynu-1(e1003)* mutant animals, there is no detectable RQ (*Figure 3c*). This suggests that the amine group on RQ derives from anthranilate or 3 HA, or some closely related product of kynureninase. To determine whether the key precursor is anthranilate, 3 HA, or both, we examined quinones in the *afmd-1(tm4547)* and *kmo-1(tm4529)* mutant strains. These contain loss-of-function mutations in either *afmd-1,* which encodes a kynurenine formamidase, or in *kmo-1* which encodes a kynurenine 3-monoxygenase. Based on current models for the kynurenine pathway, *afmd-1(tm4547)* strain cannot make either anthranilate or 3 HA whereas the *kmo-1 (tm4529)* can generate anthranilate but not 3 HA. We find that both strains have reduced RQ levels compared with N2 (*Figure 3d*) suggesting that the key precursor is 3 HA. Although the reduction in RQ levels in these two mutants is less severe than in the *kynu-1(e1003)* strain, we note that while *kynu-1* encodes the sole kynureninase in the *C. elegans* genome, there are paralogues of both *afmd-1* and *kmo-1* (*afmd-2* and *kmo-2* respectively) which may be functionally redundant. Furthermore, several of these enzymes act on multiple related substrates, which may permit additional routes of 3 HA synthesis that are not shown. If 3 HA is indeed the key precursor for RQ, it should be a substrate for the polyprenyltransferase COQ-2 — COQ-2 prenylates PHB in the first step of UQ synthesis and our model suggests it should prenylate 3 HA as the first step in RQ synthesis. Consistent with this, we identify a m/z peak corresponding to the predicted mass of prenylated 3 HA (Supp *Figure 2*), but find no equivalent peak for prenylated anthranilate (data not shown). Both the genetic data and mass spec data thus show that RQ synthesis requires the kynurenine pathway to generate 3 HA and that 3 HA is a substrate for COQ-2.

As an additional confirmation that the amine group on RQ ultimately derives from tryptophan we tested whether a tryptophan-derived aromatic amino group is being incorporated into RQ. We fed *C. elegans* 15N-labelled bacteria for three generations either in the presence or absence of 14N tryptophan. As shown in *Figure 4a*, the sole source of any 14N incorporated into RQ is the 14N tryptophan. As expected, the RQ detected in animals fed with 15N bacteria alone is approximately all 15N labelled (*Figure 4b*). However, if 14N tryptophan was added while worms are feeding on the 15N-labelled bacteria,~50% of the RQ observed was 14N RQ — the sole source for this 14N was the added tryptophan (*Figure 4b*). We note that if we add 14N 3 HA instead of 14N Trp we also detect significant 14N RQ, but see no 14N RQ if 14N anthranilate is added, further confirming that 3 HA is the key RQ precursor (*Figure 4c*). Taken together, our data suggest that RQ does not derive from UQ, and that 3 HA is the source of the amine group on the quinone ring of RQ. We propose that the pathway of RQ and UQ synthesis are largely the same — the key difference is the presence or absence of the amine group on the initial aromatic ring substrate for COQ-2.

## RQ is required for long-term survival of *C. elegans* in anaerobic conditions

As shown in *Figure 2*, *C. elegans* shows similar changes in metabolism when it is treated with KCN as STHs undergo when they adapt to the hypoxic environment of their host. For example, the classic hallmark of this RQ-dependent anaerobic metabolism is the generation of succinate by the action of Complex II as a fumarate reductase and *C. elegans* shows high levels of succinate when treated with KCN (*Figure 3b*). To confirm that this generation of succinate is indeed RQ-dependent in

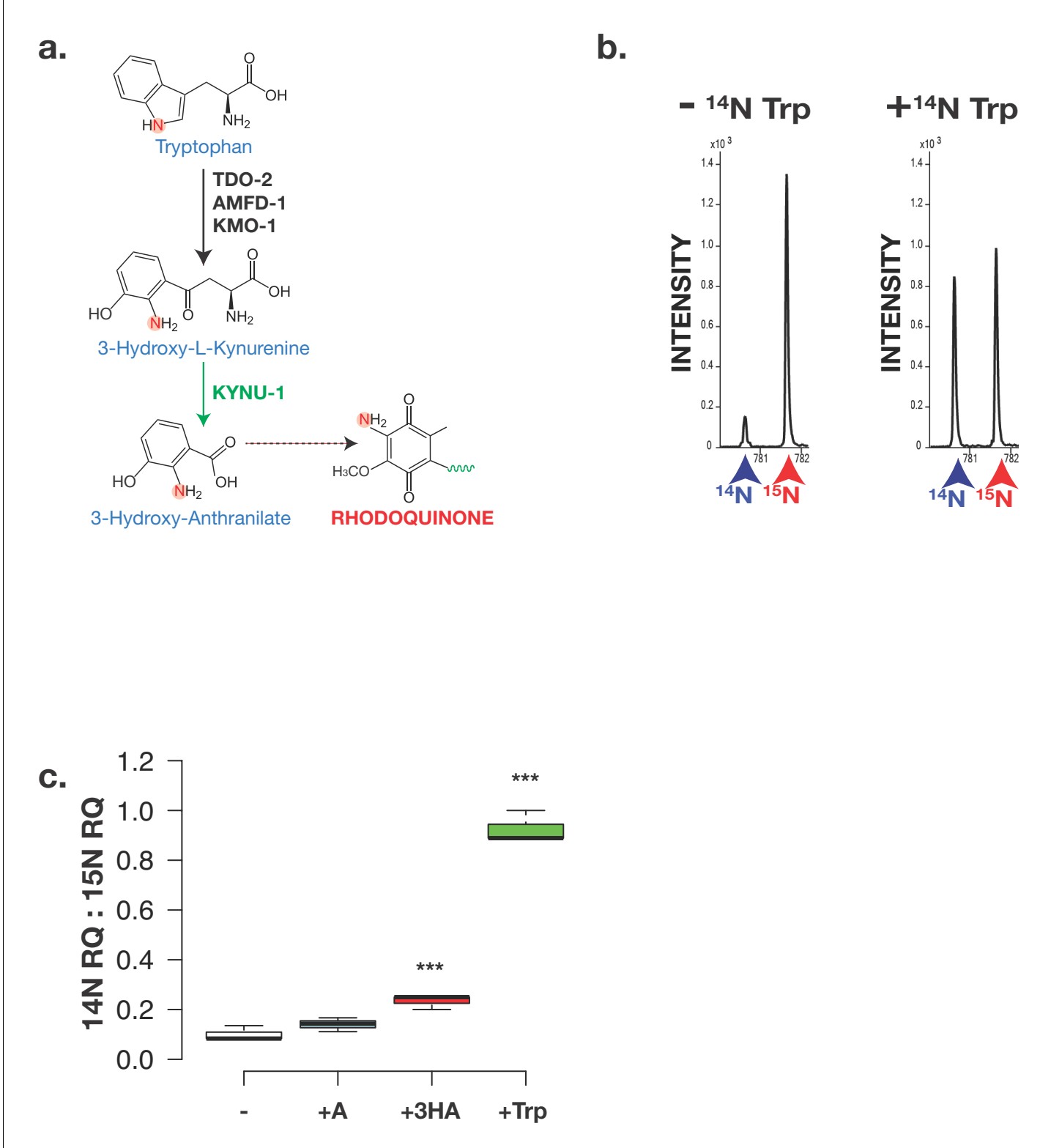

**Figure 4.** The critical amine group on RQ derives from Tryptophan. (a) Schematic of kynurenine pathway. 3-hydroxyanthranilate (3 HA) derive from Tryptophan via the kynurenine pathway and this requires KYNU-1 activity. The nitrogen atom that becomes part of the key amine group on RQ is highlighted. (b) Analysis of RQ in animals fed either only 15N substrates or 15N substrates along with 14N Tryptophan. Wild-type worms were fed with 15N-labelled bacteria for three generations either in the absence or in the presence of 14N Tryptophan. Almost all RQ is 15N labelled when worms were only eating 15N bacteria. However when 14N Tryptophan was also present, almost half the RQ is 14N RQ, indicating that the amine group of RQ

*Figure 4 continued on next page*

*Figure 4 continued*

must derive from Tryptophan. (c) Wild-type worms were fed with 15N-labelled bacteria for three generations either in the absence (-) or in the presence of 14N anthranilate (+A), 14N 3-hydroxyanthranilate (+HA) or 14N Tryptophan (+Trp). Quinones were extracted and 14N RQ and +1 15N RQ levels quantified. Graph shows ratio of 14N RQ to 15N; data are from three independent biological replicates, *** denotes p<0.01 compared to – control treatment.

DOI: https://doi.org/10.7554/eLife.48165.008

*C. elegans*, we examined whether RQ-deficient *kynu-1(e1003)* mutant animals could generate succinate when exposed to KCN. While wild-type worms generate high levels of succinate when Complex IV is inhibited with KCN, RQ-deficient *kynu-1(e1003)* mutant animals do not (*Figure 5*), confirming that the metabolic shift we see when we expose *C. elegans* to KCN is not simply similar to that of parasitic helminths in their hosts, it also requires RQ.

We have thus established that treating *C. elegans* with KCN drives them into an alternative metabolic state where they use RQ to drive the same anaerobic metabolism used by STHs in their hosts. To be able to screen efficiently for drugs that affect RQ synthesis or RQ-dependent metabolism, however, we need a direct phenotypic readout for RQ-utilisation rather than a molecular readout (such as succinate generation). When do *C. elegans* require RQ-dependent metabolism and what are the consequences if they have no RQ? Since RQ-dependent metabolism is being used in the presence of KCN we compared the sensitivity of wild-type worms and *kynu-1(e1003)* mutants to KCN and the ability of wild-type worms and *kynu-1(e1003)* mutants to survive in KCN for long periods. We found no significant differences in acute KCN sensitivity of wild-type worms and *kynu-1 (e1003)* mutants — both slow their movement in the presence of KCN, stop moving completely by ~90 min (*Figure 6a*), and remain immobile from there on when maintained in KCN. However, there was a dramatic difference in their ability to survive extended periods in KCN. We exposed worms to KCN for different lengths of time and then removed animals from KCN and assayed their movement over the next 3 hr as they recover from KCN treatment. When wild-type worms are removed from KCN, they rapidly recover movement (*Figure 6b*; recovery is also shown as a movie in *Video 1*) — they can do this even after 24 hr of KCN treatment (*Figure 6c*). However, *kynu-1(e1003)* mutants show greatly reduced ability to survive extended KCN treatment — they do not survive exposure to KCN for 12 hr or more (*Figure 6c*). We confirmed this using a second mutant strain that also carries a loss-of-function mutation in *kynu-1*, the *kynu-1 (tm4924)* strain and find essentially identical results (data not shown). RQ-dependent anaerobic metabolism thus allows *C. elegans* to survive extended periods where it cannot use oxygen as the terminal electron acceptor of the ETC.

This provides a simple high throughput assay for drugs that specifically affect RQ-dependent metabolism: drugs that block RQ synthesis or the activity of RQ-dependent pathways should abolish the ability of worms to survive >12 hr in KCN. To test this, we used the compound wact-11 — this is a Complex II inhibitor that binds to the quinone-binding pocket of Complex II (*Burns et al., 2015*). wact-11 is highly related to the anthelmintic flutolanil (*Burns et al., 2015*)

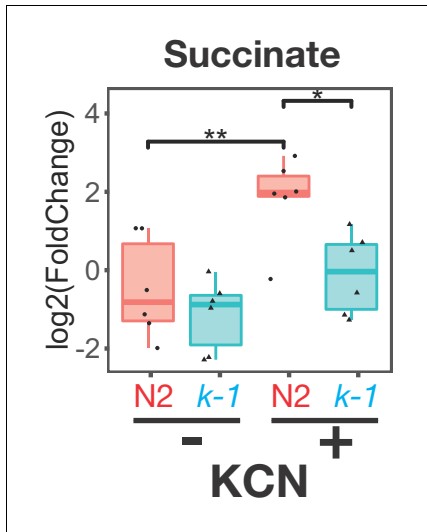

**Figure 5.** Buildup of succinate following Complex IV inhibition requires RQ. Wild-type (N2) or *kynu-1(e1003)* mutant animals were exposed to 200 µM KCN for 6 hr and metabolites extracted and analysed by mass spectrometry. Data are from five independent biological repeats; box plots show median and the interquartile range as well as individual data points; levels are computed relative to levels in wild-type untreated animals. Succinate levels increase markedly in N2 worms (red 'N2') following KCN treatment; there is no significant increase in *kynu-1(e1003)* mutant animals (cyan 'k-1') confirming that fumarate reductase activity of Complex II requires RQ.

DOI: https://doi.org/10.7554/eLife.48165.009

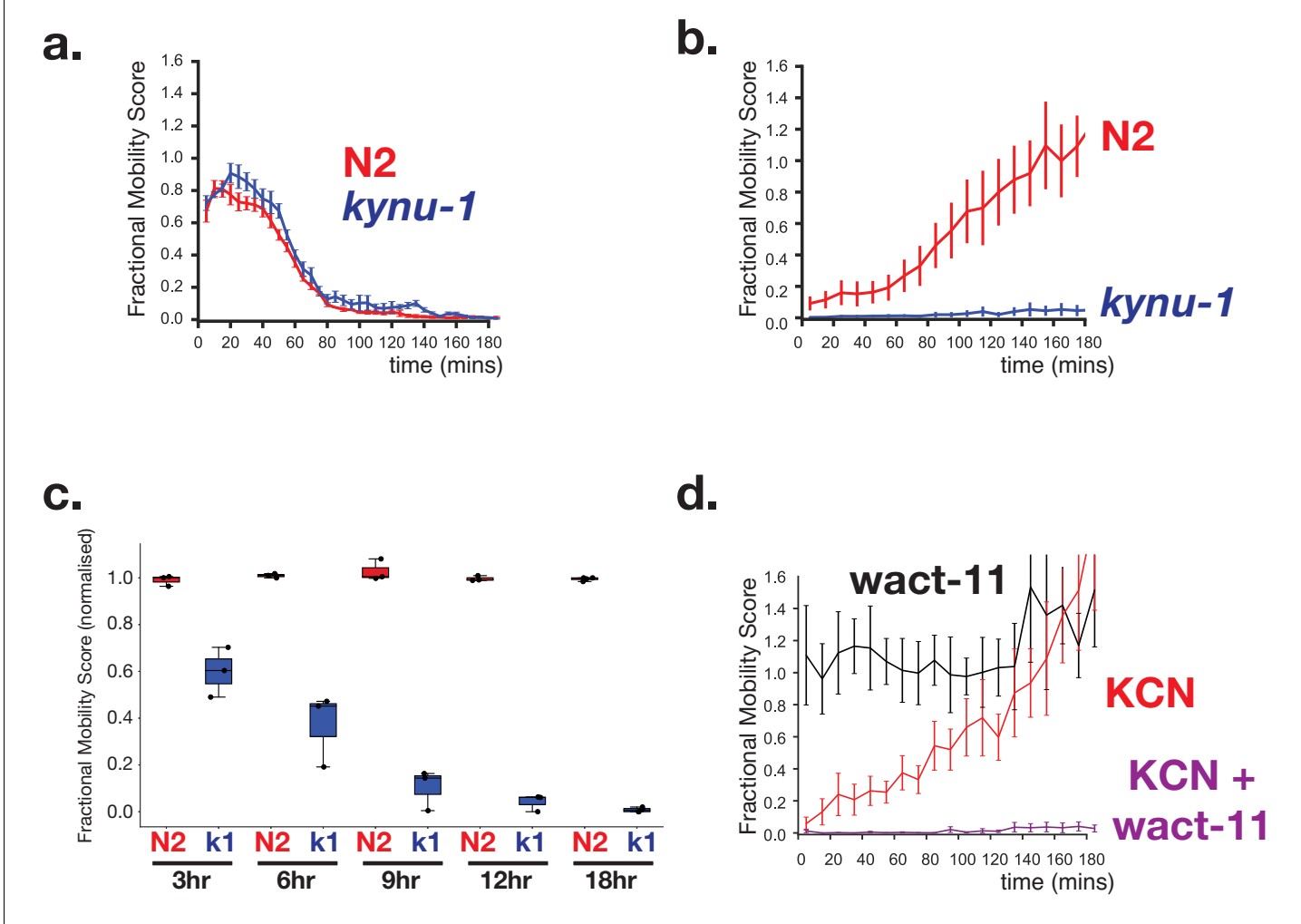

**Figure 6.** RQ-dependent metabolism is required for long-term survival in anaerobic conditions. (a) Loss of RQ does not affect acute sensitivity to KCN. Wild-type (N2, red curve) and *kynu-1(e1003)* mutant L1 animals (*kynu-1,* blue curve) were exposed to 200 µM KCN and their movement measured over 3 hr (see Materials and methods). Both strains slow their movement and become immobile after ~90 mins. Curves show means of 3 biological replicates with three technical replicates in each; error bars are standard error. (b) RQ is required for survival following extended treatment with KCN. Wild-type (N2, red curve) and *kynu-1(e1003)* mutant L1 animals (*kynu-1,* blue curve) were exposed to 200 µM KCN for 15 hr. KCN was then diluted (see Materials and methods) and worm movement was measured over a 3 hr time course. Curves show means of 3 biological replicates with three technical replicates in each; error bars are standard error. (c) Effect of different lengths of exposure to KCN on worm survival. Wild-type (N2, red) and *kynu-1 (e1003)* mutant L1 animals (*kynu-1,* blue) were exposed to 200 µM KCN for different lengths of time from 3 hr to 18 hr. KCN was then diluted (see Materials and methods) and worm movement was measured after 3 hr. Box plots show levels of movement after a 3 hr recovery period. Data are from three biological replicates with three technical replicates in each. (d) Ability to survive extended KCN exposure requires Complex II activity. Wild-type L1 animals were treated with 200 µM alone, 10 µM of wact-11 (a helminth-specific Complex II inhibitor) alone, or a combination of KCN and wact-11 for 15 hr. Drugs were then diluted 6x and worm movement was measured over a 3 hr timecourse. wact-11 treatment alone had no effect on survival over the experiment (black curve) and worms recovered completely from KCN treatment alone. However worms could not survive treatment with both KCN and wact-11. Curves show means of 3 biological replicates with three technical replicates in each; error bars are standard error.
DOI: https://doi.org/10.7554/eLife.48165.010

and is highly selective for helminth Complex II (*Burns et al., 2015*). Complex II is critical for RQ-dependent anaerobic metabolism where it acts as a RQ-dependent fumarate reductase — inhibitors of Complex II would thus be expected to prevent survival in KCN. When worms are treated with wact-11 alone, we see no impact on worm movement over the time course of our assay. However, when worms are treated with both wact-11 and KCN, they can no longer survive long-term KCN exposure (*Figure 6d*) indicating that this assay can allow discovery of drugs that block RQ-dependent anaerobic metabolism. Our assay should thus allow efficient screens for drugs that inhibit RQ

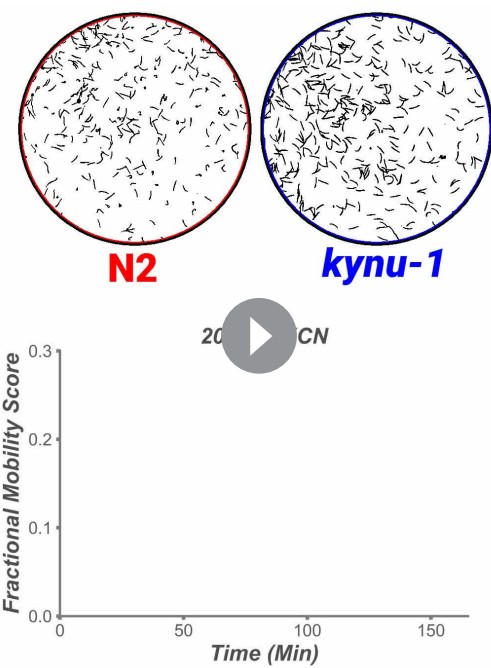

**Video 1.** Survival in KCN of N2 but not of *kynu-1 (tm4924)* mutants. L1 animals were treated with 200 µM KCN for 18 hr followed by a 6-fold dilution. Worms were imaged using the Phenalysys Parallux 2. Graphs show the moving average of raw FMS values as described in Materials and methods.
DOI: https://doi.org/10.7554/eLife.48165.011

synthesis or RQ utilization in vivo in a helminth under conditions where they require RQ, the first time this has been possible. The assay is image-based and quantitative and can be extremely high throughput when using rapid imaging platforms like the Phenalysys Parallux 2 (used to make the movies of recovery in *Video 1*) which can measure the effect of 96 drugs on worm movement in under 5s. We note that not all drugs that kill *C. elegans* when in the presence of KCN (i.e. that would be hits in this assay) will affect RQ synthesis or RQ use and secondary screens will be required to further stratify the hits. Nonetheless, inhibitors of RQ synthesis or use can be discovered using this assay and we anticipate that this will yield novel compounds that may be effective anthelmintics.

Finally, we took advantage of a set of mutant worm strains that are resistant to wact-11 treatment. Mutations that result in resistance to wact-11 treatment cluster in the quinone binding pocket of Complex II (*Burns et al., 2015*) and we reasoned that some of these might not only prevent the binding of wact-11 but might fortuitously also disrupt the binding of RQ and thus affect the ability of worms to survive extended exposure to KCN. We tested a number of point mutants that affect wact-11 sensitivity and found that most mutants appear similar to wild-type worms in their ability to survive long term KCN exposure (*Figure 7a* and data not shown). However, we found that the G71E mutation results in worms that are unable to survive extended KCN exposure — the G71E animals thus resemble *kynu-1(e1003)* mutants. We note that this mutation sits right above the modelled binding site for the rhodoquinone ring, whereas a neighbouring mutation that sits two turns of an alpha-helix further away (C78Y) has no effect. We thus suggest that the G71E mutation affects the ability of *C. elegans* to bind RQ into the quinone binding pocket of Complex II and thus to drive RQ-dependent fumarate reduction as part of its RQ-dependent anaerobic metabolism.

## Discussion

RQ was first identified over 50 years ago (*Moore and Folkers, 1965*). It is absolutely required for the anaerobic metabolism used by parasitic helminths to survive in the hypoxic environment of the host gut where they can thrive for many months. The single amine group that differs between RQ and UQ is crucial for this — it allows RQ to carry electrons of the right electropotential to drive quinone-coupled dehydrogenases (QDHs) in reverse, acting as reductases (*Fioravanti and Kim, 1988*; *Sato et al., 1972*). In aerobic conditions, QDHs carry electrons from a diverse set of electron donors and transfer them onto UQ and hence into the ETC; under anaerobic conditions, RQ carries electrons to the QDHs which then reduce a diverse set of electron sinks, providing an exit point for electrons from the ETC. The single amine group on the quinone ring of RQ allows parasites to carry out this unusual anaerobic metabolism and thus it affects the lives of over a billion humans. Despite the importance of RQ for human health, its synthesis has been elusive and no anthelmintics have been identified that affect RQ synthesis. Here, we used *C. elegans* genetics to identify the source of the key amine group on RQ, and to establish a pipeline for screening for new compounds that alter the ability of worms to make and use RQ.

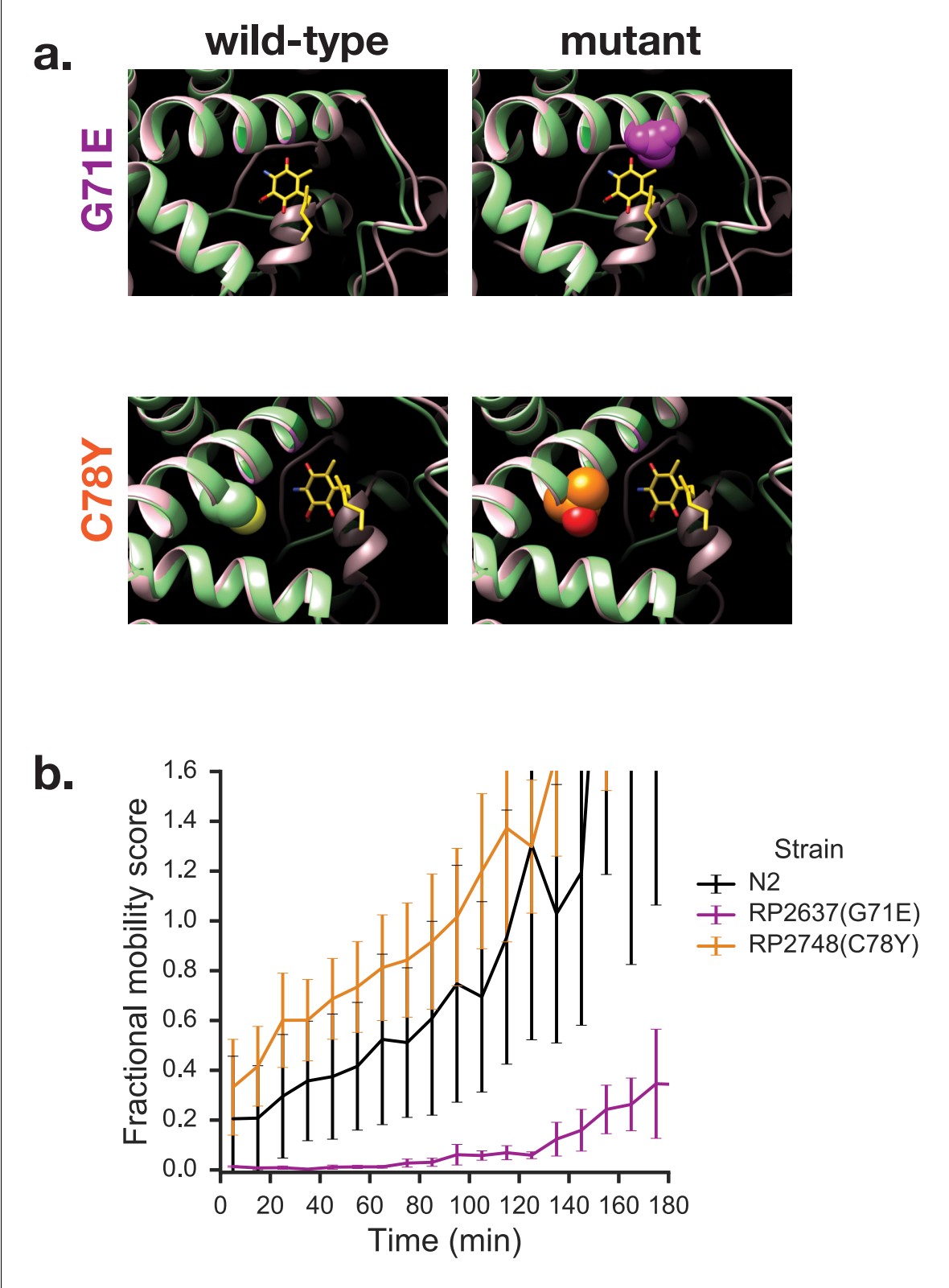

**Figure 7.** Mutations in quinone-binding pocket of Complex II subunit MEV-1 affect ability to survive extended KCN treatment. (a) Positions of mutations that alter wact-11 sensitivity. MEV-1 structures show either the wild-type residues or mutations that alter wact-11 binding. The structures shown are *C. elegans* sequences (green) threaded onto the *Ascaris suum* crystal (pink). RQ is shown in yellow with the critical amine group in blue — note that this is RQ2 and not RQ9, hence the shortened prenyl tail. Note that G71E alters a residue that lies right over the quinone ring of RQ whereas

*Figure 7 continued on next page*

*Figure 7 continued*

C78Y is further from the ring. (**b**) Effect of quinone-pocket mutations on ability to survive extended KCN exposure. L1 worms containing either wild-type, G71E or C78Y mutant *mev-1* alleles were exposed to 200 µM KCN for 15 hr. KCN was then diluted 6x and the movement of worms measured across a 3 hr time course. Curves show means of 3 biological replicates with three technical replicates in each; error bars are standard error.

DOI: https://doi.org/10.7554/eLife.48165.012

The critical question in RQ synthesis is where the amine group on the quinone ring comes from and how it is added. Previous studies from bacteria suggested that RQ synthesis uses UQ as a precursor (*Brajcich et al., 2010*) and that the amine group is added at a late stage in RQ synthesis. At least in bacteria, this seems to be the case — in *R.rubrum*, the gene *rquA* that is required for RQ synthesis encodes a methyltransferase that is related to the quinone methyltransferases UbiG/COQ3 and UbiE/COQ5 that act in UQ synthesis (*Lonjers et al., 2012*; *Stairs et al., 2018*). Very recently, RquA has been shown to be able to catalyse the conversion of UQ to RQ (*Buceta et al., 2019*)— in bacteria, RQ synthesis thus requires UQ synthesis and the key amine group is a late stage addition. Here, however, we show that the mechanism of RQ synthesis in *C. elegans* is different. In *C. elegans* (and likely in helminths in general) RQ synthesis does not require UQ as a precursor and, crucially, the critical amine group on the quinone ring of RQ is not added at a late stage in the synthesis of RQ but is present from the initial steps of RQ synthesis. We show that the amine group likely comes from 3-hydroxyanthranilate (3 HA), which is generated by degradation of tryptophan via the kynurenine pathway. We also demonstrate that RQ synthesis requires the kynurenine pathway. We note that during revision of the manuscript, a second group also found that RQ synthesis requires the kynurenine pathway (*Buceta et al., 2019*). Thus, for the first time since its discovery in the early 1960s, we now have a key insight into how RQ is made in helminths and this has several implications for the search for novel anthelmintics that might affect RQ synthesis.

First, we do not believe that there are completely separate dedicated pathways for UQ and for RQ synthesis in helminths. Instead, we suggest that UQ and RQ have a largely shared synthesis pathway. A key difference in RQ and UQ synthesis is the use of different initial substrates for the polyprenyltransferase COQ-2. If PHB is used, the product will be UQ; if 3 HA (or possibly a related product of tryptophan metabolism) is used, the product will be RQ. This 'different precursor, shared pathway' model for RQ and UQ synthesis stands in contrast to the pathways used by bacteria to synthesise two other quinones, UQ and menaquinone (MK). Facultative anaerobic bacteria including non-pathogenic *E. coli* as well as major human pathogens like *M. tuberculosis* make two quinones: UQ which is used as an electron carrier under aerobic conditions, and menaquinone (MK) (reviewed in *Meganathan and Kwon, 2009*), which acts as a carrier under anaerobic conditions. The synthesis pathways of UQ and MK are completely distinct and the genes involved are distinct (reviewed in *Meganathan and Kwon, 2009*). This separation of MK and UQ synthesis pathways has allowed the development of a number of promising compounds that act as specific inhibitors of MK synthesis (*Kurosu et al., 2007* and *Debnath et al., 2012*; reviewed in *Boersch et al., 2018*). We suggest that there may be no analogous inhibitors for 'the RQ synthesis pathway' in helminths since there does not appear to be a dedicated RQ pathway analogous to the dedicated MK synthesis pathway. This also raises intriguing questions of regulation of quinone content in helminths. While in bacteria the switch from UQ (aerobic) to MK (anaerobic) occurs through the use of two entirely separate biosynthetic pathways, in helminths the switch from UQ (aerobic) to RQ (anaerobic) must occur via the entry of different substrates into a largely shared pathway. We currently have little insight into how that switch occurs or how it is regulated.

Second, the pathway we identify for RQ synthesis suggests novel targets for anthelmintics. The finding that the kynurenine pathway is the source of the key precursors for RQ synthesis suggests naively that helminth-specific inhibitors of the kynurenine pathway might act as potent anthelmintics. However, the human gut is likely to be a source of anthranilate and 3 HA from host metabolism or from the microbiome and inhibiting production of these molecules in the helminth itself might thus prove ineffective. A more likely target is COQ-2, the enzyme that prenylates the amine-containing 3 HA ring as the first step in RQ synthesis. While host and parasite COQ-2 are orthologous enzymes, they have different substrate specificity: the host only makes UQ and not RQ, whereas the parasite COQ-2 must be able to use not only PHB for UQ synthesis but also amine-containing precursors like

3 HA for RQ synthesis. The ability of helminth COQ-2 to use amine-containing substrates efficiently thus opens up the possibility of helminth-specific COQ-2 inhibitors, a potential avenue for new anthelmintics. We believe that the in vivo assay we describe here may help identify such drugs.

Our study also raises key new questions. One of the most intriguing to us is why RQ synthesis is so rare amongst animals. To date, only three groups of animals are known to make RQ: molluscs, annelids, and helminths (*Allen, 1973*; *Fioravanti and Kim, 1988*; *Sato and Ozawa, 1969*; *Takamiya et al., 1999*; *Van Hellemond et al., 1995*). If RQ synthesis and UQ synthesis largely share a common pathway, why doesn't every animal that makes UQ also make RQ? One possibility is that while some of the pathway for UQ and RQ synthesis is shared, RQ synthesis requires additional components that might only be present in RQ-synthesising species. For example, we note that while the UQ precursor PHB has a hydroxyl group at the four position, the RQ precursor 3 HA does not and this must be added by some as yet unknown enzyme. The other possibility is that key enzymes in the UQ and RQ synthesis pathway may have altered specificity in species that make RQ to allow them to use substrates containing an amine group for RQ synthesis as well as non-aminated substrates for UQ synthesis. Careful phylogenetic analysis may identify subtle sequence changes that could allow helminth enzymes to use amine-containing RQ precursors. Such molecular signatures that mark out RQ-utilising species have been found in proteins that bind both UQ and RQ — for example there are helminth-specific residues that lie around the quinone binding site in Complex II (*Burns et al., 2015*). It is clear that there is no simple answer yet for the finding that RQ is made by so few animals but that this will likely emerge as more of the RQ synthesis pathway is uncovered.

Finally, we note that several steps in the kynurenine pathway and in the ubiquinone synthesis pathway are catalysed by either monooxygenases or dioxygenases that require oxygen. These include TDO-2 (*Hayaishi et al., 1957*) and KMO-1 (*Detmer and Massey, 1985*; *Entsch et al., 1976*) in the kynurenine pathway and COQ-6 (*Ozeir et al., 2015*) and CLK-1 (*Marbois and Clarke, 1996*) in the UQ synthesis pathway. RQ synthesis thus appears to require the availability of oxygen for these enzymes, an unexpected result since RQ is preferentially required in anaerobic conditions and is the predominant quinone in helminths living under anaerobic conditions. How might these oxygen-requiring steps be carried out for RQ synthesis? It is possible that the helminth enzymes have evolved so that they can still operate under low oxygen conditions. Other oxygen-using proteins have evolved extremely high oxygen affinity in helminths — for example *Ascaris* haem is octameric and binds oxygen with ~25,000 times greater affinity than human haem (*Minning et al., 1999*). Alternatively, these same enzymatic steps might be carried out by other enzymes in lower oxygen conditions. In *E. coli*, for example, in aerobic conditions UbiB and UbiF carry out the same hydroxylation modifications to the quinone ring as *C. elegans* COQ-6 and CLK-1 and mutation of either gene results in a lack of mature UQ in these bacteria (reviewed in *Meganathan and Kwon, 2009*). However, under anaerobic conditions, *ubiB* and *ubiF* mutants make normal levels of UQ suggesting that other enzymes carry out these reactions in low oxygen conditions (*Alexander and Young, 1978*). It is possible that there is an analogous set of enzymes that are required for RQ synthesis in low oxygen conditions — these would carry out similar reactions to COQ-6 and CLK-1 but without the requirement for oxygen. If they exist, they remain to be discovered.

There is thus still much to be discovered about the regulation and the precise pathway of RQ synthesis in helminths. The results presented here provide a firm starting point and the assay we describe for drugs that affect RQ-dependent metabolism may lead to the discovery and development of a new class of anthelmintic drugs. Since resistance to known classes of anthelmintics is widespread among livestock parasites like *H.contortus*, *C.oncophora* and *A.suum* and is rising in human populations (reviewed in *Sangster et al., 2018*), this will prove critical in the control and treatment of these major pathogens.

## Materials and methods

### Worm strains and maintenance

In addition to the traditional laboratory strain N2, here we include work using strains *clk-1(qm30)*, *kynu-1(e1003)*, *sdhc-1(tr357)*, and *sdhc-1(tr423)*. The two *sdhc-1* strains were provided by Dr. Peter Roy and all other strains were provided by the *Caenorhabditis* Genetics Centre or by the Mitani

group of the National Bioresources Project. All worms were maintained on NGM agar plates seeded with *E. coli* OP50 as described elsewhere (*Stiernagle, 2006*) and maintained at 20℃.

## RQ tryptophan Anti-Labelling experiment

*Escherichia coli* (MG1655) was grown overnight at 37℃ in M9 media prepared using 1 g/L [15]N ammonium chloride (Cambridge Isotopes) as the nitrogen source. Bacteria were heat killed at 65℃ for 15 min. The heated culture was used to seed NGM agar plates. 500 µL of 50 mg/ml tryptophan in water was spread on each 10 cm plate. Ten L4 nematodes were placed on each plate to lay eggs overnight at 20℃. Adult worms were removed following the egg laying period. After 5 days at 20℃ the nematodes were collected and frozen at −80℃.

## Quinone extraction

Nematode samples were thawed and lysed via sonication. Quinone extraction solvent containing a 2:1 ratio of chloroform and methanol (Thermo Fisher Optima LC-MS grade) respectively was added to the samples. The organic phase of the sample was collected and then dried using nitrogen gas. Samples were resuspended in a 60:40 acetonitrile and isopropanol solution prior to analysis using APCI LC-MS.

## Quinone LC-MS analysis

Quinones were analyzed by reverse phase chromatography on an Eclipse Plus C-18 RRHD column, 2.1 mM x 50 mm with 1.8 um packing operated in a thermostatted column compartment held at 70℃. Buffer A was 50% MeCN in water, Buffer B was 100% acetone with 0.01% formic acid. Starting conditions were 0.25 mL/min at 50% B. Gradient was 1 min hold, followed by increase 100% B at 5 min, hold 100% B until 7 min, then return to 50% B at 7.1 min and hold until 10 min. Samples were introduced from a HTC pal by injection of 5 µL sample into a 2 µL loop. Wash one was acetonitrile and wash two was isopropanol. Samples were ionised using a Multimode ionisation source (Agilent) operated in APCI mode, gas temp 350℃, vaporizer temp 350℃, drying gas 5 L/min, nebulizer 60 PSI, capillary voltage 4000 V, corona current 4 µA, skimmer voltage 70 V, octupole 1 RF 400 V. Samples were analyzed on a 6230 TOF, a 6545 Q-TOF, or a 6490 QQQ as indicated. Fragmentor voltage for TOF/QTOF analysis was 200 V. For QQQ analysis, ubiquinone nine was monitored by MRM of 795.6/197.3 at CID of 52 V; rhodoquinone-9 was monitored at 780.6/192.1 at CID of 52 V.

## Image-based assays

All image-based experiments were conducted on L1 animals which were collected from mixed-stage plates and isolated using a 96 well 11 µM Multiscreen Nylon Mesh filter plate (Millipore: S5EJ008M04) as described previously (*Spensley et al., 2018*) to a final concentration of ~100 animals per well. They were then incubated in a final concentration of 200 µM potassium cyanide for varying amounts of time. There are two key assays: the acute assay and the recovery assay. The acute assay monitors worm movement immediately following exposure to KCN every 5 min for a total of 3 hr. The recovery assays involve a KCN incubation of 3, 6, 9, 12, or 18 hr after which the KCN is diluted 6-fold with M9 buffer. Immediately after dilution, worm movement is monitored every 10 min for 3 hr. In both assays, worm movement is quantified using an image-based system as previously described (*Spensley et al., 2018*). All data were normalised by the fractional mobility score of the M9-only control wells per strain per time point.

For movies shown in *Video 1*, N2 and *kynu-1(tm4924)* L1 worms were incubated in 200 µM KCN for 18 hr as per the recovery assay. Images were taken and processed using a Phenalysys Parallux 2. Raw images were taken every second and segmented into a set of images per individual well. These are then processed by FFmpeg to generate a Quicktime movie. The accompanying graph plots the moving average of total-well FMS scores using a box filter with n = 60, with every datapoint representing a minute of the assay.

## Drug preparation and assay assembly

Solutions of potassium cyanide (Sigma 60178–25G) were made fresh prior to each experiment in phosphate buffered saline (PBS) and then diluted to a 5 mM stock solution in M9 buffer. 2X working concentrations were then prepared with M9 and the KCN stock solution. Wact-11 (Chembridge ID

6222549) was kept frozen as a 100 mM stock in DMSO. Diluted wact-11 stocks were made in DMSO (BioShop DMS666) to a concentration of 3.75 mM and kept frozen until day of use. 10X working concentrations were made with wact-11 stock, M9 buffer, and DMSO. All experiments were prepared to contain 0.8% v/v DMSO to control for any confounding effects of drug solvent.

Assays were assembled in flat-bottomed polystyrene 96-well plates (Corning 3997) to a total volume of 100 μL and 40 μL for the acute and recovery assays, respectively. Apart from assays including wact-11 which constituted half KCN solution, 10% wact-11 solution, and 40% worms in buffer, all other assays were comprised of equal parts worms in buffer and KCN solution.

### Rotenone LC-MS

Worms were collected and isolated as described above. A final concentration of 7.5 L1s/10 μL was treated with final concentrations of 12.5 μL rotenone (Sigma R8875) in 0.8% DMSO and with 0.8% DMSO alone and with 100 μM KCN in M9 or with M9 alone, 20 mL altogether in 40 mL plastic containers (Blender Bottle 600271) and were on a shaker for 1 hr at room temperature. After 1 hr, samples were poured over 0.2 μM Nylaflo nylon filter membranes (PALL 66604) over vacuum and once the supernatant had run through, the filter paper was placed in prepared 1.2 mL of 8:1:1 extraction solvent (MeOH, HPLC Grade (SA 34860); $H_2O$, HPLC Grade (Caledon 8801-7-40); $CHCl_3$, HPLC Grade (SA 650498)) in a 1.5 mL microfuge tube on dry ice. Tubes were inverted five times then vortexed. Samples were switched between −80°C and −20°C three times. Filters were then removed, and the tubes spun at 13,200 rpm at 4°C for 30 min. 1 mL of supernatant was transferred to a new tube and dried under dry $N_2$ with <0.02% $O_2$ at 5 PSI for 8 hr. Each sample was reconstituted with 30 μL HPLC grade water as was prepared labelled yeast reference. Samples and reference were spun at 13,200 rpm at 4°C for 5 min. 10 μL sample and reference were placed in an LC-MS sample vial (Agilent 5190–2243, cap is Agilent 5185–5820) and were fast spun at 1,000 rpm at 4°C.

### *kynu-1* LC-MS

N2 and *kynu-1(e1003)* worms were washed and filtered as previously described. They were then placed in 1.5 mL microfuge tubes at 1.5 mL for a final concentration of 45 L1s, 300 μM KCN in M9 or M9 alone and were placed on a rotator for 6 hr at room temperature. After 6 hr, the samples were extracted and prepared as previously described.

### Succinate analysis

Succinate was extracted from an ion-paired reverse phase method IPRP method LC-MS run at an mzCenter of 117.0193 and a retention time of 690 s in the case of the *kynu-1* experiment and from an Acid method LC-MS run at an mzCenter of 117.0193 and a retention time of 141 s in the case of the rotenone experiment. Both sets of samples were normalised to a labelled yeast reference. In the case of the *kynu-1* experiment, they were further normalised to the median of all the extracted peaks for each sample. In both cases they were ultimately normalised to the mean unlabelled N2 sample treated with buffer or 0.8% DMSO. Plots were generated using {plotPeak}.

### Structural analysis

Mitochondrial rhodoquinol-fumarate reductase from *A. suum* bound with rhodoquinone-2 (PDB: 3VR8) was displayed on Chimera (*Pettersen et al., 2004*) and the *C. elegans* sequence was threaded by homology using Modeller (*Sali and Blundell, 1993*; *Webb and Sali, 2016*) and the MSA with 15 iterations.

## Acknowledgements

The research in this study was supported by CIHR grant 501584. We thank Prof. Peter Roy and multiple members of Fraser and Roy labs for insightful discussions, Prof. Brent Derry for intellectual stimulation, the *Caenorhabditis* Genetics Center and the group of Shohei Mitani for *C. elegans* strains, Olga Zaslaver and Angela Wong for their time and help with mass spec analyses. Structural graphics and analyses performed with UCSF Chimera, developed by the Resource for Biocomputing, Visualisation, and Informatics at the University of California, San Francisco, with support from NIH P41-GM103311.

## Additional information

### Competing interests

Mark A Spensley: Affiliated with Phenalysys Inc. The author has no other competing interests to declare. The other authors declare that no competing interests exist.

### Funding

| Funder | Grant reference number | Author |
| --- | --- | --- |
| Canadian Institutes of Health Research | 501584 | Andrew G Fraser |

The funders had no role in study design, data collection and interpretation, or the decision to submit the work for publication.

### Author contributions

Samantha Del Borrello, Investigation, Methodology, Writing—review and editing; Margot Lautens, Data curation, Formal analysis, Investigation, Writing—review and editing; Kathleen Dolan, Investigation, Writing—review and editing; June H Tan, Formal analysis, Investigation, Methodology, Writing—review and editing; Taylor Davie, Formal analysis, Visualization; Michael R Schertzberg, Data curation, Investigation, Writing—review and editing; Mark A Spensley, Conceptualization, Resources, Software, Formal analysis, Writing—review and editing; Amy A Caudy, Conceptualization, Supervision, Funding acquisition, Investigation, Methodology, Project administration, Writing—review and editing; Andrew G Fraser, Conceptualization, Supervision, Funding acquisition, Writing—original draft, Project administration

### Author ORCIDs

Samantha Del Borrello  https://orcid.org/0000-0002-0117-0592
Margot Lautens  http://orcid.org/0000-0002-8503-9603
June H Tan  http://orcid.org/0000-0001-6597-3952
Mark A Spensley  https://orcid.org/0000-0001-6167-4461
Amy A Caudy  https://orcid.org/0000-0001-6307-8137
Andrew G Fraser  https://orcid.org/0000-0001-9939-6014

### Decision letter and Author response

Decision letter https://doi.org/10.7554/eLife.48165.015
Author response https://doi.org/10.7554/eLife.48165.016

## Additional files

### Supplementary files

• Transparent reporting form
DOI: https://doi.org/10.7554/eLife.48165.013

### Data availability

All data in manuscript and supporting files.

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
