## [Decision Letter]

Thank you for submitting your article "Identification of the pathway of Rhodoquinone biosynthesis in *C. elegans*" for consideration by *eLife*. Your article has been reviewed by three peer reviewers, and the evaluation has been overseen by a Reviewing Editor and Gisela Storz as the Senior Editor. The following individual involved in review of your submission has agreed to reveal their identity: William F (Bill) Martin (Reviewer #1).

The reviewers have discussed the reviews with one another and the Reviewing Editor has drafted this decision to help you prepare a revised submission.

In this manuscript, Del Borrello et al. investigate the biosynthesis of rhodoquinone (RQ) in *C. elegans*. RQ is an electron carrier very similar in structure to coenzyme Q (CoQ, aka ubiquinone (UQ)), but that is found in very few animals. The fact that it exists in parasitic helminths, which require RQ for survival under low oxygen tension, makes the RQ pathway a potential target for therapeutics; however, the pathway of RQ biosynthesis is largely unknown. The authors confirm that RQ is made in *C. elegans*, and that it does not rely on the activity of clk-1 (coq7), the enzyme that performs the penultimate step of CoQ biosynthesis. The main contribution of this paper is that the RQ synthesis in *C. elegans* requires the gene kynu-1, which encodes an enzyme in the kynurenine pathway, which produces anthranilate from tryptophan. Overall, this well-written manuscript adds a solid new piece to our understanding of RQ biosynthesis, and sets up the potential to target the RQ pathway therapeutically. The reviewers were enthusiastic about the importance of the work, but found some of the claims overstated based on the data provided. As outlined below, if additional data are provided and/or some of the claims toned down (or better supported), this work would be suitable for publication in *eLife*.

Essential revisions:

1) The clearest and most important conclusion from this paper is that kynureninase can produce an RQ precursor from tryptophan (presumably 3-Hydroxyanthranilic acid (3-HA), although this is not shown). This single finding, while interesting and important, is the sole basis of the claim that they have "identified the pathway of RQ biosynthesis" – a major overstatement, as is the title of the paper. That is, there is much left to work out before one can claim that the RQ pathway is known: The work does not show that 3-HA is the direct precursor, that known upstream (TDO2, AMFD-1) or downstream (where does the 4-hydroxyl come from?) enzymes in the kynurenine pathway are required, that the known CoQ enzymes (e.g., Coq3, 5, 6) are used, or that DMQ couldn't also be a source of RQ (they eliminated this as a possibility just by stating that it would be thermodynamically unfavorable, even though Figure 4 suggests that other routes to RQ are possible). There is simply no need for the authors to overstate their findings. The impact of the paper would be strengthened if any of the persistent gaps above were addressed.

2) At a minimum, the authors should provide additional evidence to shore up the claim that anthraniliate-derived 3-HA is the precursor used by Coq2. For example, it is possible that the amine group derived from tryptophan via the kynurenine pathway is incorporated into the tyrosine-derived CoQ precursor, 4-hydroxybenzoate. It seems that the authors' model is right, but this can be tested by inhibition of Hfd1, the enzyme that generates the CoQ precursor 4-hydroxybenzoate (4-HB) (Stefely et al. Nat Biotech 2016, Payet et al. Cell Chem Bio 2016), and by showing evidence of polyprenyl-hydroxyanthranilate formation using their same HPLC assays.

3) The model the authors propose is that 3-HA can be used by Coq2 because of its tolerance of substitutions at the 5 and 6 positions of 4-HB, which is a reasonable claim. However, the kynurenine pathway exists in many animals that don't have RQ. Why would this likewise not result in RQ formation, if this is the differentiating point, as seems to be claimed? In other words, what is special about helminths (and molluscs) that allows them to make RQ? The authors discuss this point and suggest that it might be a set of quinone methyltransferases that are present in *C. elegans*, all parasitic helminths that they have checked (i.e., other nematodes), as well as in molluscs and annelids. Since they are discussing "helminths" in this paper, it would be reasonable for them also to include flatworms (including F. hepatic and S. mansoni, both of which have been reported to make and use RQ – see below) in this analysis. Because flatworms, annelids, and molluscs represent a different assemblage of invertebrates than do nematodes, it would be interesting to consider whether they have come up with the same or different solutions for making RQ. The authors should consider moving their description of these results and the figure in which they are presented to the Results section instead of the Discussion.

4) The authors claim that drugs that abolish nematode survival in KCN provide a simple assay for drugs that specifically affect RQ- dependent metabolism. This system can indeed find such drugs, but there are many RQ-independent ways to kill nematodes during KCN treatment, so it is not specific. The authors even give an example: "wact-11" is a complex II inhibitor that kills nematodes during KCN treatment without changing RQ levels. Although blocking complex II is related to RQ function, it is hard to see how such an approach would be a promising way to treat helminth infections in humans (which also rely on complex II). For the screening method to be effective, it would need to be more RQ dependent, less dependent on general mitochondria health. Again, the claim here is too strong.

---

## [Author Response]

Essential revisions:1) The clearest and most important conclusion from this paper is that kynureninase can produce an RQ precursor from tryptophan (presumably 3-Hydroxyanthranilic acid (3-HA), although this is not shown). This single finding, while interesting and important, is the sole basis of the claim that they have "identified the pathway of RQ biosynthesis" – a major overstatement, as is the title of the paper. That is, there is much left to work out before one can claim that the RQ pathway is known: The work does not show that 3-HA is the direct precursor, that known upstream (TDO2, AMFD-1) or downstream (where does the 4-hydroxyl come from?) enzymes in the kynurenine pathway are required, that the known CoQ enzymes (e.g., Coq3, 5, 6) are used, or that DMQ couldn't also be a source of RQ (they eliminated this as a possibility just by stating that it would be thermodynamically unfavorable, even though Figure 4 suggests that other routes to RQ are possible). There is simply no need for the authors to overstate their findings. The impact of the paper would be strengthened if any of the persistent gaps above were addressed.

We agree that we were over-enthusiastic with our claims and completely agree that there is still much that is unknown about the full RQ biosynthetic pathway. We have toned down the claims at multiple positions starting with the title. We think that what we have shown is that RQ synthesis requires the kynurenine pathway (and state that throughout as the main finding rather than ‘Identification of the pathway for RQ synthesis’ as before) and focus largely on the conclusion that the key amine group on RQ does not come as a late addition in a reaction using UQ as an obligate precursor, rather it derives from 3HA.

We now provide additional data for this – we measure the RQ levels in afmd-1 and kmo-1 mutant worms and find a strong reduction in both cases. We note that the effect is less profound than in the kynu-1 mutant animals where RQ is essentially undetectable but also that there are paralogues afmd-2 and kmo-2 which may be partially redundant. We hope that this strengthens our paper and show these results now in an extra panel of Figure 3 We confirm our finding with kynu-1(e1003) mutant worms with a second allele of kynu-1 and also provide videos showing that survival in KCN requires kynu-1. We hope that taken together the reviewers agree that this strengthens the paper.

2) At a minimum, the authors should provide additional evidence to shore up the claim that anthraniliate-derived 3-HA is the precursor used by Coq2. For example, it is possible that the amine group derived from tryptophan via the kynurenine pathway is incorporated into the tyrosine-derived CoQ precursor, 4-hydroxybenzoate. It seems that the authors' model is right, but this can be tested by inhibition of Hfd1, the enzyme that generates the CoQ precursor 4-hydroxybenzoate (4-HB) (Stefely et al. Nat Biotech 2016, Payet et al. Cell Chem Bio 2016), and by showing evidence of polyprenyl-hydroxyanthranilate formation using their same HPLC assays.

This sounds trivial on the face of it but there is a large family of *C. elegans* paralogues related to Hfd1 which complicate this analysis. alh-4 and alh-5 are >85% identical and are the closest matches for Hfd1 but our experience of these large paralogue families is that it is non-trivial to predict which of the paralogues can compensate for/functional replace each other. This would therefore be a lengthy analysis and unlikely to yield a satisfying answer. We do however now provide mass-spec evidence for polyprenyl 3HA formation (Figure 3—figure supplement 1) and note that we cannot find any polyprenyl anthranilate, which I believe strongly supports the claim that 3HA is the substrate for COQ-2 for RQ synthesis. Does this seem reasonable to you as support for the model, along with the addition afmd-1 and kmo-1 RQ level data?

3) The model the authors propose is that 3-HA can be used by Coq2 because of its tolerance of substitutions at the 5 and 6 positions of 4-HB, which is a reasonable claim. However, the kynurenine pathway exists in many animals that don't have RQ. Why would this likewise not result in RQ formation, if this is the differentiating point, as seems to be claimed? In other words, what is special about helminths (and molluscs) that allows them to make RQ? The authors discuss this point and suggest that it might be a set of quinone methyltransferases that are present in C. elegans, all parasitic helminths that they have checked (i.e., other nematodes), as well as in molluscs and annelids. Since they are discussing "helminths" in this paper, it would be reasonable for them also to include flatworms (including F. hepatic and S. mansoni, both of which have been reported to make and use RQ – see below) in this analysis. Because flatworms, annelids, and molluscs represent a different assemblage of invertebrates than do nematodes, it would be interesting to consider whether they have come up with the same or different solutions for making RQ. The authors should consider moving their description of these results and the figure in which they are presented to the Results section instead of the Discussion.

After consulting the Senior Editor, we have removed this last figure to streamline the paper and to focus it specifically on the core of the results showing the requirement of the kynureninase pathway for RQ synthesis. We anticipate following the evolutionary signature of RQ synthesis more in the future, so hope to expand on these initial findings at that point and hope that this refocusing satisfies the reviewers. We note that we still mention this albeit more briefly in the Discussion section:

“One possibility is that while some of the pathway for UQ and RQ synthesis is shared, RQ synthesis requires additional components that might only be present in RQ-synthesising species. […] The other possibility is that key enzymes in the UQ and RQ synthesis pathway may have altered specificity in species that make RQ to allow them to use substrates containing an amine group for RQ synthesis as well as non-aminated substrates for UQ synthesis.”

We also do not claim any solution to this question and now state more clearly “It is clear that there is no simple answer yet for the finding that RQ is made by so few animals but that this will likely emerge as more of the RQ synthesis pathway is uncovered.”

4) The authors claim that drugs that abolish nematode survival in KCN provide a simple assay for drugs that specifically affect RQ- dependent metabolism. This system can indeed find such drugs, but there are many RQ-independent ways to kill nematodes during KCN treatment, so it is not specific. The authors even give an example: "wact-11" is a complex II inhibitor that kills nematodes during KCN treatment without changing RQ levels. Although blocking complex II is related to RQ function, it is hard to see how such an approach would be a promising way to treat helminth infections in humans (which also rely on complex II). For the screening method to be effective, it would need to be more RQ dependent, less dependent on general mitochondria health. Again, the claim here is too strong.

I am not sure we fully agree with this. We claim that our assay allows us “to screen efficiently for drugs that affect RQ synthesis or RQ-dependent metabolism” and that our assay reports RQ-utilization. We do not claim that every drug that comes back from such a primary screen will do that – of course there will be other ways to kill worms that have been treated with KCN and this drug screen like all others will need secondary assays. Nonetheless drugs that affect RQ synthesis or that specifically block RQ docking to the quinone binding pockets of quinone-coupled dehydrogenases should work as anthelmintics and should emerge as hits from our screen. To support this we give the example of wact-11. The reviewers’ state that “Although blocking complex II is related to RQ function, it is hard to see how such an approach would be a promising way to treat helminth infections in humans (which also rely on complex II)” but seem to have missed that flutolanil (the example we choose) is exactly that – an anthelmintic that is a Complex II inhibitor.(“wact-11 is highly related to the anthelmintic flutolanil (Burns et al., 2015) and is highly selective for helminth Complex II (Burns et al., 2015). Complex II is critical for RQ-dependent anaerobic metabolism where it acts as a fumarate reductase inhibitors of Complex II might thus alter survival in KCN.”) It works because Complex II in helminths has diverged from human Complex II – many of the residues surrounding the quinone binding site in helminth Complex II are helminth-specific and this is thought to be due to the requirement for binding of both UQ and RQ to helminth Complex II whereas human Complex II only has to dock UQ. Why does flutolanil (or wact-11) kill *C. elegans* in the presence of KCN? It isn’t because of a general effect on ‘mitochondrial health’ (at least it has no phenotype in our assay conditions) but likely because it blocks the docking of RQ to Complex II and thus blocks the ability of Complex II to accept electrons from RQ and act as a fumarate reductase.

We now add a caveat: “We note that not all drugs that kill *C. elegans* when in the presence of KCN (i.e. that would be hits in our assay) will affect RQ synthesis or RQ use and secondary screens will be required to further stratify the hits. Nonetheless, inhibitors of RQ synthesis or use can be discovered using this assay and we anticipate that this will yield novel compounds that may be effective anthelmintics.” We hope this satisfies the reviewers.